# The effect of human amnion epithelial cells on lung development and inflammation in preterm lambs exposed to antenatal inflammation

**Paris Clarice Papagianis**[1,2,3,4]*, **Siavash Ahmadi-Noorbakhsh**[3], **Rebecca Lim**[1,2], **Euan Wallace**[1,2], **Graeme Polglase**[1,2], **J. Jane Pillow**[3☯‡], **Timothy J. Moss**[1,2☯‡]

1 The Ritchie Centre, Hudson Institute of Medical Research, Clayton, Victoria, Australia, 2 Department of Obstetrics and Gynaecology, School of Clinical Health Sciences, Monash University, Clayton, Victoria, Australia, 3 School of Human Sciences, The University of Western Australia, Crawley, WA, Australia, 4 School of Health Sciences and Health Innovations Research Institute, RMIT University, Melbourne, VIC, Australia

☯ These authors contributed equally to this work.
‡ These authors are joint senior authors on this work.
* pariscpapagianis@gmail.com

**Data Availability Statement:** All relevant data are within the manuscript and its Supporting information files.

## Abstract

### Background

Lung inflammation and impaired alveolarization are hallmarks of bronchopulmonary dysplasia (BPD). We hypothesize that human amnion epithelial cells (hAECs) are anti-inflammatory and reduce lung injury in preterm lambs born after antenatal exposure to inflammation.

### Methods

Pregnant ewes received either intra-amniotic lipopolysaccharide (LPS, from *E.coli* 055:B5; 4mg) or saline (Sal) on day 126 of gestation. Lambs were delivered by cesarean section at 128 d gestation (term ~150 d). Lambs received intravenous hAECs (LPS/hAECs: n = 7; $30 \times 10^6$ cells) or equivalent volumes of saline (LPS/Sal, n = 10; or Sal/Sal, n = 9) immediately after birth. Respiratory support was gradually de-escalated, aimed at early weaning from mechanical ventilation towards unassisted respiration. Lung tissue was collected 1 week after birth. Lung morphology was assessed and mRNA levels for inflammatory mediators were measured.

### Results

Respiratory support required by LPS/hAEC lambs was not different to Sal/Sal or LPS/Sal lambs. Lung tissue:airspace ratio was lower in the LPS/Sal compared to Sal/Sal lambs (P<0.05), but not LPS/hAEC lambs. LPS/hAEC lambs tended to have increased septation in their lungs versus LPS/Sal (P = 0.08). Expression of inflammatory cytokines was highest in LPS/hAECs lambs.

**Funding:** This research was supported by an NHMRC Project Grant (1077769), NHMRC Centre for Research Excellence (1057514), two NHMRC Senior Research Fellowships (JJP; 1077691: TJM 1043294), the Victorian Government's Operational Infrastructure Support Program, and the West Australian Government's Medical and Health Research Infrastructure Fund. Unrestricted equipment and consumable support was provided by Chiesi Farmaceutici S.p.A. (poractant alfa); Fisher & Paykel Healthcare (ventilator circuits); and ICU Medical (arterial monitoring lines). None of the commercial industry funders had any input into study design, data collection and analysis, decision to publish or preparation of the manuscript. Chiesi Farmaceutici S.p.A. reviewed the final manuscript for technical accuracy pertaining to description and use of their surfactant, in accordance with a Material Transfer Agreement associated with the provision of surfactant for study animals. The University of Western Australia (via JJP) has consultancy agreements with Chiesi Farmaceutici S.p.A. unrelated to the subject of this study. Fisher & Paykel Healthcare have material transfer agreements with Hudson Research Institute that are also unrelated to this work. There are no other relevant interests relating to employment, consultancy, patents, or products in development or marketed products to declare. These material transfer agreements do not alter our adherence to PLOS ONE policies on sharing data and materials. The funders had no role in study design, data collection and analysis, decision to publish, or preparation of the manuscript.

**Competing interests:** The University of Western Australia (via JJP) has consultancy agreements with Chiesi Farmaceutici S.p.A. unrelated to the subject of this study. Fisher & Paykel Healthcare have material transfer agreements with Hudson Research Institute that are also unrelated to this work. There are no other relevant interests relating to employment, consultancy, patents, or products in development or marketed products to declare. These material transfer agreements do not alter our adherence to PLOS ONE policies on sharing data and materials. None of the authors have conflicts of interest to disclose.

## Conclusions

Postnatal administration of a single dose of hAECs stimulates a pulmonary immune response without changing ventilator requirements in preterm lambs born after intrauterine inflammation.

## Introduction

Antenatal inflammation (manifest as histological chorioamnionitis) often precedes preterm birth and may increase the likelihood of bronchopulmonary dysplasia (BPD) [1]. Preterm infants often require respiratory support, which may exacerbate lung inflammation and contribute to BPD development. Preterm infants exposed to chorioamnionitis or postnatal mechanical ventilation show evidence of airway inflammation, including elevated levels of interleukin (IL)-1, IL-6 and IL-8 in tracheal aspirates [2–5]. The combination of chorioamnionitis and postnatal ventilation increases the risk of BPD [2]. Thus, prevention or attenuation of postnatal lung inflammation in response to chorioamnionitis and mechanical ventilation may provide a realistic therapy for BPD [6].

Human amnion epithelial cells (hAEC) are a potential anti-inflammatory therapy for BPD [7], with several potential benefits: stem cell-like hAECs can differentiate along any cell lineage, including lung epithelial cells *in vitro* [8] and *in vivo* [9]; reduce T cell proliferation [10]; and promote macrophage polarisation to the M2 pro-regenerative phenotype [11–15]. Lung inflammation is reduced by hAECs in mice exposed to bleomycin [16], and in fetal sheep exposed to intra-amniotic lipopolysaccharide (LPS) [12] or ventilation *in utero* [11, 17]. Most of the existing studies on the effects of hAECs in perinatology provided proof-of-principle of therapeutic effect, but utilized animal models of overt inflammation and injury which do not mimic human clinical scenarios. The effects of hAECs in newborns exposed to both prenatal inflammation and postnatal ventilation are unknown, and some fundamental aspects about the use of hAECs remain unknown.

The lungs of preterm lambs exposed to antenatal inflammation [18] and/or mechanical ventilation have simplified architecture with large airspaces and attenuated formation of secondary septae, reflecting inhibition of alveolarization [19], consistent with the pulmonary structure of infants who died from BPD [20, 21].

We hypothesised that a pro-inflammatory antenatal stimulus would result in reduced lung function and impaired lung development in the first week after preterm delivery, and that administration of hAECs would prevent this adverse outcome. Furthermore, we hypothesized that hAEC function is unaffected by temperature ranges expected to be encountered clinically.

## Materials and methods

See the online supplement for a detailed description of methods (Supplemental Methodology available via private sharing link: https://doi.org/10.6084/m9.figshare.14527386.v1).

### Ethics

Procedures were approved by Monash University Human Ethics Committee (MUHREC-CF13/2144-2013001109). The University of Western Australia Animal Ethics Committee approved all animal studies (RA 3/100/1301 and RA 3/100/1454) which were performed in

accordance with the Australian code for the care and use of animals for scientific purposes
[22].

**Isolation, cryopreservation and preparation of hAECs.**    Placentae were obtained from women undergoing elective term cesarean section, after the provision of written informed consent. The isolation of hAECs from placentae is outlined elsewhere [8]. Live-cell counts and viability were determined by trypan blue exclusion ($>$ 85% viability was required) before use. Isolates of hAECs from at least 2 donors were thawed, combined, counted and resuspended in sterile saline for administration to preterm lambs and subsequent *in vitro* studies.

## Preterm lamb studies

**Antenatal interventions.**    Pregnant ewes received intramuscular (IM) medroxyprogesterone acetate (150 mg; Pfizer, Australia) at 120 days' gestational age (GA; term ~150 d). Ultrasound-guided intra-amniotic (IA) injection of lipopolysaccharide (LPS;4 mg; 2 mg/mL; *Escherichia coli* 055:B5; Sigma-Aldrich, NSW, Australia; n = 10) or saline (n = 9) was performed at 126 days' GA. This dose of LPS results in a well characterized fetal inflammatory response that peaks 48 h after injection [23]. Ewes received an intramuscular (IM) injection of betamethasone (5.7 mg/dose; Celestone, Merck Sharp & Dohme Pty Ltd, Australia) 48 h and 24 h prior to planned cesarean section delivery at 128 days' gestational age (GA; term ~150 d), commencing 6 hours after IA LPS injection.

**Preterm delivery.**    Anaesthetized ewes underwent hysterotomy at 128 d gestation. The externalized fetus was intubated and lung liquid was drained passively. Exogenous surfactant (3 mL, 80 mg/mL poractant alfa, Chiesi Farmaceutici S.p.A., Italy) was administered via the endotracheal tube. The lamb was dried, delivered and weighed and suspended in ventral recumbency within a sling, on a neonatal baby warmer (Fisher& Paykel Healthcare). The lamb received a sustained inflation (30 cmH$_2$O for 30 s, F$_I$O$_2$ 0.30) before initiation of ventilation. The ewe was euthanized immediately after delivery (150 mg/kg pentobarbitone; Valabarb, Jurox, Australia).

**Postnatal interventions.**    Preterm lambs exposed to IA LPS (n = 10) received either hAEC (LPS/hAEC; 30 x 10$^6$ hAECs in 20 mL saline IV at 1 mL/min; n = 7) or an equivalent volume of saline (n = 9) commencing immediately after delivery. Controls received IA saline (2 mL) antenatally and 20 mL IV saline postnatally.

**Respiratory support.**    Lambs received graded respiratory support and ventilation (Evita® Infinity® V500 Ventilator, Dräger) in accordance with human clinical practice guidelines, beginning with mechanical ventilation (MV), then bubble continuous positive airway pressure (B-CPAP), followed by extubation onto heated humidified high flow (HHF) via nasal cannulae and eventual unassisted breathing of room air. MV was initiated with volume guarantee (5–7 mL/kg), a fraction of inspired oxygen (F$_I$O$_2$) of 0.3, peak inspiratory pressure (PIP) of 30 cmH$_2$O, positive end-expiratory pressure (PEEP) of 9 cmH$_2$O and ventilator rate of 50 breaths/min. Ventilator adjustments were determined by clinical examination and arterial blood gas measurements by a neonatologist (JJP); therefore respiratory care of lambs was adjusted as needed and not necessarily sequentially administered. The target peripheral oxyhaemoglobin saturation (SpO$_2$) was 90–95% at the lowest achievable F$_I$O$_2$. Target partial pressure of arterial carbon dioxide (PaCO$_2$) was 40–50 mmHg.

Pulmonary gas exchange was assessed using arterial partial pressure of oxygen (PaO$_2$) to F$_I$O$_2$ (P/F) ratio and oxygenation index (OI = [MAP x F$_I$O$_2$ x 100] / [PaO$_2$ x 1.36]) where mean airway pressure (MAP) is in cmH$_2$O and PaO$_2$ is in mmHg.

**General postnatal management.**    Detailed general postnatal management of lambs is outlined in the Supplementary Methods including how physiological parameters were measured

for each type of respiratory support and observations including, hourly body temperature by rectal probe, heart rate (HR), blood pressure (BP), respiratory rate (RR), arterial oxyhaemoglobin saturation ($SaO_2$) by pulse oximetry, mean airway pressure (MAP) and $F_IO_2$ were recorded.

Arterial blood gases were obtained every 30 minutes for the first hour, every hour for the next 4 h and as required thereafter.

## Tissue collection

On day 7, lambs were killed (sodium pentobarbitone 150 mg/kg). The left lung was inflation-fixed at 30 $cmH_2O$ with 10% formaldehyde [24] for morphometric and histological analyses. Static lung compliance was calculated as the volume of fixative infused per kilogram body weight divided by the inflation pressure (30 $cmH_2O$).

**Histological analyses.** Fifteen lung sections per animal were stained with Harts resorcin-fuschin [25] for identification of elastin, tissue, airspace and septal crests. A semi-automated ImageJ (NIH image, Bethasda, Maryland, USA) plugin (Copyright © 2015, Keith Schulze, Monash Micro Imaging, Monash University) was used to detect tissue, airspace and septae. Haematoxylin and eosin-stained sections were used for scoring epithelial sloughing (indicative of mechanical disruption of the lung epithelium) [26, 27] using a scale between 0 and 4: no events = 0; < 5 events = 0.5; 5–10 events = 1–2; 10–20 events = 2.5–3; and > 20 events = 3.5–4. The birefringence of collagen was visualized with picrosirus red (PSR) stain using a polarizing microscope and normalised to area of tissue [28].

Immunohistochemistry was used to identify leukocytes (CD45) [29], macrophages (CD163) [30], proliferating cells (Ki67) [19], myofibroblasts (α-smooth muscle actin; (SMA)) [19], and type II alveolar epithelial cells (pro-surfactant protein (SP)-C) [31]. All sections were scanned using ImageScope (Aperio Technologies, USA) and analysed using ImageJ, except α-SMA density which was analysed using ImagePro Plus software (c. 9.2, Build 6156, 2012–2015 Media Cybernetic© Inc.), as described elsewhere [19].

**Gene expression analyses.** Total RNA was extracted from subpleural segments of the right lower lobe of the lung and the midline of the liver. mRNA levels for genes of interest (S1 Table) were measured using TaqMan® probes.

Briefly, expression of each target gene was measured in triplicate cDNA samples on an ABI Prism 7900HT Real-Time PCR System (PE Applied Biosystems). mRNA levels were calculated from Ct values using qBase+ software (v3.1) [32]. Reference gene ribosomal (r)18S was selected using the qBase+ geNorm algorithm for assessing stability of reference genes [33]. All genes were expressed as calibrated normalized to relative quantity (CNRQ; i.e. normalized to r18S and expressed relative to the Sal/Sal group), where r18S Sal/Sal is ~0.00. The use of CNRQ is a modification of delta-delta CT method for analysing PCR which takes into account multiple reference genes and gene specific amplification, inter-run calibration and proper error propagation, as outlined in Hellemans et al. [35].

## Effect of thermal environment on wound-healing properties of hAECs

In a complementary study, the functional viability and wound healing capacity of hAECs was determined at different temperatures to establish if different thermoneutral environments of human and ovine cells could impact on the use of hAECs in the preterm lamb.

Isolates of ureaplasma free hAECs [34] were seeded at 5 x $10^5$ cells/well on a 6-well plate and incubated at 33°C, 37°C or 39°C. Cells were cultured in standard DMEM/F12 media (Gibco, Life Technologies) with 10% FBS and 1% Penicillin-Streptomycin (Gibco, Life Technologies), in 5% $CO_2$ in room air. Cells were harvested at 24, 48 or 72 hours and stained with

Annexin V and 7AAD (PE Annexin V Apoptosis Kit I, BD Biosciences, USA) to assess apoptotic activity [35]. Data were acquired with BD FACS-Canto II flow cytometer. Conditioned medium was collected from hAEC cultures at 72 hours.

Wound-healing assays were performed on hAECs as described previously [36]. Cells were incubated at 37°C until confluent (7–12 days) and a scratch was made across the well. After scratching, hAECs were incubated at 33°C, 37°C or 39°C for 72 h. Recovery of the scratched area was analysed using ImageJ [37].

Immortalized mouse macrophages (iMACs; provided by A/Prof Ashley Mansell, Hudson Institute of Medical Research) were plated at 5 x $10^5$ cells/well in a 96-well plate. The next day, medium was removed and replaced with 100 μL of FITC-marked fluorescent beads (2 x $10^4$ beads/μL; diameter: 1 μm; Sigma-Aldrich, Fluka, USA) in media and 100 μL of hAEC-conditioned media (from 72 h cultures at 33°C, 37°C or 39°C) and incubated for 3 h at 37°C to allow phagocytosis/uptake of beads. Negative controls were iMACs cultured in 200 μL media. Positive controls were iMACs cultured in 100 μL media with 100 μL fluorescent beads. Data were acquired and analysed using a BD FACS-Canto II flow cytometer. Uptake was expressed as a percentage of fluorescent cells, corresponding to the percentage of iMACs with phagocytosed beads.

## Statistics

*In vitro* data were analysed using 2-way ANOVA with Holm-Sidak's multiple comparison test (for apoptosis) or one-way ANOVA with Tukey's post hoc test (for wound healing and phagocytosis). For *in vivo* data, missing physiological data points ($< 5\%$) were imputed using sequential regression modelling (SPSS v. 24, IBM) prior to analysis with two-way repeated measures ANCOVA. Pearson's correlations were used to identify covariates for ANCOVA. Year of delivery was used as a covariate in all physiological analyses. Additional covariates are stated where included. Post-hoc comparisons were made using the Holm-Sidak method. Histological data are expressed as mean ± SD and were analysed using Kruskal-Wallis with Dunn's multiple comparisons post-hoc test (GraphPad Prism v7). TaqMan® data are expressed as mean ± SD.

## Results

### Preterm lamb studies

**Characteristics of lambs.** Lamb characteristics, including gestational age and birthweight, were comparable between treatments and are summarized in Table 1.

**Respiratory support.** $PaCO_2$, $HCO_3^-$ and base excess increased in all lambs over the 7-day period indicative of compensated respiratory acidosis (P<0.05; Fig 1). $PaO_2$ was lower on day 1 in LPS/Sal (61.2 ± 11.1 mmHg; P = 0.05) and LPS/hAEC (57.1 ± 12.8 mmHg; P = 0.030) lambs, compared to Sal/Sal lambs (85.1 ± 29.0 mmHg), but there were no inter-group differences in $PaO_2$ between days 2–7 (Fig 1). LPS/Sal lambs required increasing $F_IO_2$ over the study (F(5, 136) = 2.5; P = 0.03), but there were no differences in $F_IO_2$ requirements between groups. $SaO_2$, pF ratio and OI values were not different between groups, with the exception of higher OI (i.e. worse oxygenation) in LPS/hAEC lambs compared to Sal/Sal lambs on day 6 (P = 0.02).

Duration of mechanical ventilation correlated with year of delivery and birth-weight ($r = 0.602$, P<0.01 and $r = 0.398$, P<0.001, respectively). Ventilation requirements were not different between Sal/Sal and LPS/Sal groups (Table 2). LPS/hAEC lambs spent a similar proportion of time on MV compared to LPS/Sal lambs, but a higher proportion of time on MV

**Table 1. Lamb characteristics.**

| Variable | Experimental group | | |
|---|---|---|---|
| | **Sal/Sal** | **LPS/Sal** | **LPS/hAEC** |
| GA (d) | 129.7 ± 1.4 | 128.9 ± 0.9 | 126.6 ± 0.79 |
| N (% male) | 9 (44.4) | 10 (50.0) | 7 (42.9) |
| Birth body weight (kg) | 2.89 ± 0.14 | 3.29 ± 0.13 | 3.12 ± 0.8 |
| PM body weight (kg) | 2.78 ± 0.13 | 3.02 ± 0.09 | 2.90 ± 0.05 |
| Left lung weight/PM body weight (g/kg) | 11.5 ± 1.2 | 17.9 ± 1.9* | 17.0 ± 1.8* |
| Static lung compliance ((mL/kg)/cmH$_2$O) | 1.97 ± 0.87 | 0.76 ± 0.05 | 0.53 ± 0.06* |
| Liver weight/PM body weight (g/kg) | 33.4 ± 2.3 | 33.6 ± 2.0 | 35.7 ± 2.6 |
| Thymus weight/PM body weight (g/kg) | 1.96 ± 0.37 | 2.37 ± 0.56 | 2.96 ± 1.13 |
| Spleen weight/PM body weight (g/kg) | 1.90 ± 0.32 | 2.38 ± 0.18 | 2.67 ± 0.70 |
| Adrenal weight/PM body weight (g/kg) | 0.28 ± 0.02 | 0.22 ± 0.05 | 0.22 ± 0.05 |

Data are mean ± SD. PM = post mortem; GA = gestational age.

*P<0.05 significantly different to Sal/Sal.

compared to Sal/Sal lambs (F(2, 135) = 3.0; P = 0.05; Table 2; accounting for year of delivery and birth-weight as cofactors).

Total hours of intubation and the proportion of time spent on ET-CPAP, nB-CPAP, nHHF or no respiratory support were not different between groups (Fig 2). There were no differences in mean airway pressure between Sal/Sal, LPS/Sal and LPS/hAEC lambs (Table 2).

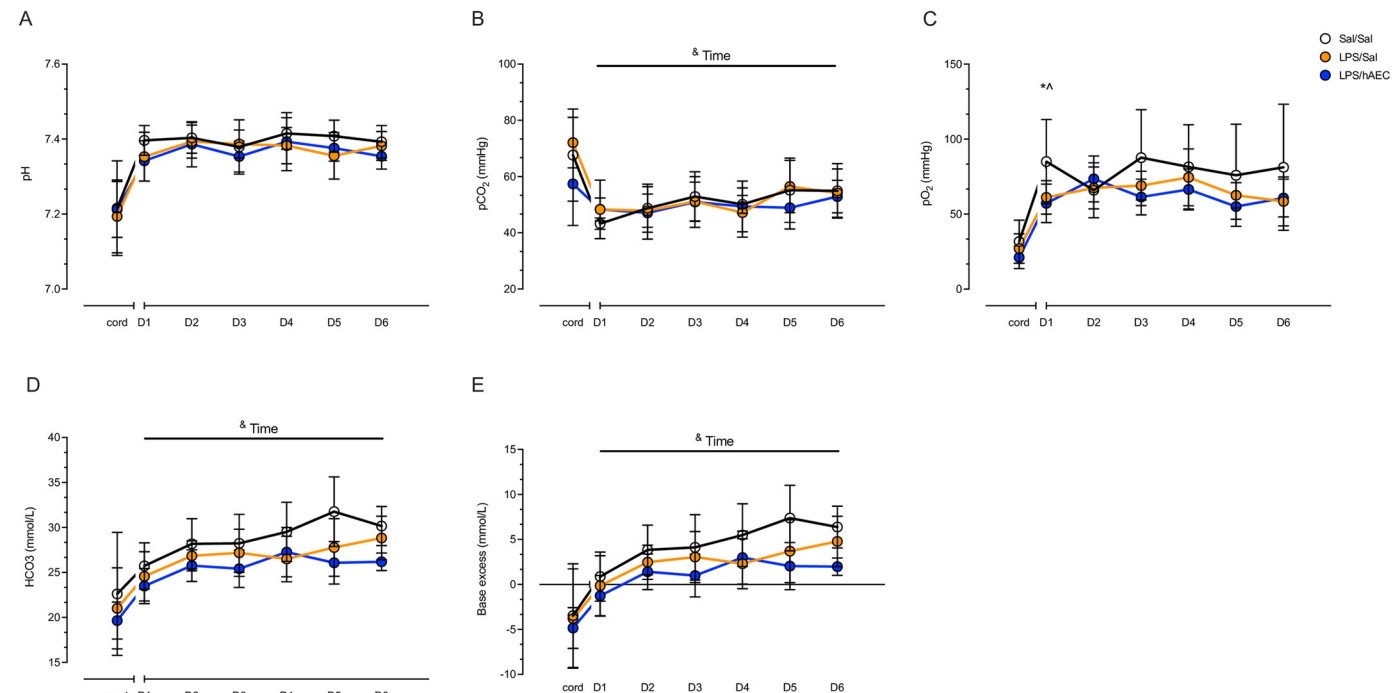

**Fig 1. pH, PaCO2, PaO2, HCO3 and BE of Sal/Sal lambs (open circles), LPS/Sal lambs (orange circles) and LPS/hAEC lambs (blue circles).** Cord blood gases were analysed separately to postnatal blood gases days 1–6 (D1-D6). ^Signifies P<0.05 between Sal/Sal and LPS/Sal. *Signifies P<0.05 between Sal/Sal and LPS/hAECs &Time signifies P<0.05 change over 1–6 days, not between treatment groups. Data are mean ± SD.

**Table 2.  Mechanical ventilation, time intubated and mean airway pressure in preterm lambs.**

|  |  | Mean difference [95% CI] | P-value |
|---|---|---|---|
| **MV (%)** |  |  |  |
| Sal/Sal | LPS/Sal | -2.63 [-6.92, 1.67] | 0.367 |
|  | LPS/hAEC | -6.33 [-12.64, -0.03] | 0.049 |
| **Time intubated (h)** |  |  |  |
| Sal/Sal | LPS/Sal | -2.64 [-6.78, 1.51] | 0.333 |
|  | LPS/hAEC | -5.14 [-11.27, 1.00] | 0.129 |
| **MAP (mmHg)** |  |  |  |
| Sal/Sal | LPS/Sal | -1.19 [-2.55, 1.95] | 0.145 |
|  | LPS/hAEC | -0.30 [-2.66, 0.27] | 0.984 |

Mean difference between Sal/Sal and LPS/Sal or LPS/hAEC lambs and [95% confidence intervals]. Year of delivery was used as a covariate in all analyses, with additional covariates of birth-weight for MV analysis and MV for MAP analyses. MV: mechanical ventilation; MAP: mean airway pressure. MAP is taken from ventilator and set continuous positive airway pressure.

**Physiology.**   Physiological variables are shown in Table 3. Mean BP, HR and RR were not different between groups, except on day 4 when HR was higher in Sal/Sal lambs than in LPS/Sal lambs (P = 0.04) and on day 6 when RR was higher in Sal/Sal lambs compared to in LPS/hAEC (P = 0.03).

Lung-to-body-weight ratio was lowest in Sal/Sal lambs compared to LPS/Sal and LPS/hAEC lambs (P = 0.03 and P = 0.04, respectively). Static lung compliance was not different in Sal/Sal lambs compared to LPS/Sal lambs, but lower in LPS/hAEC lambs compared to Sal/Sal lambs (P = 0.001).

**Lung histology and morphology.**   Lung architecture was heterogeneous between and within treatment groups (Fig 3). The proportion of tissue in the lungs was not different between Sal/Sal (47.6 ± 2.5%), LPS/Sal (55.9 ± 2.7%) or LPS/hAEC (50.3 ± 2.4%) lambs.

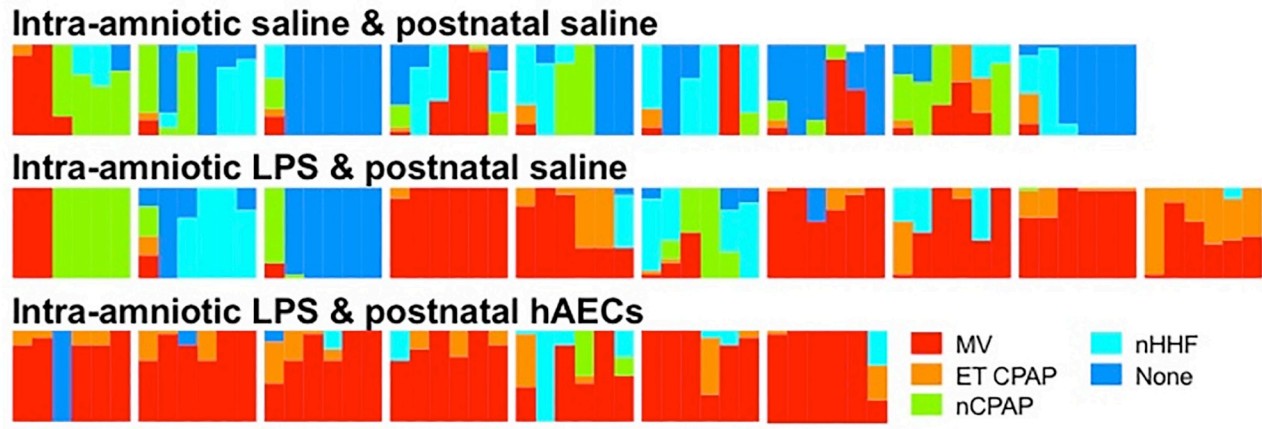

**Fig 2. Heat maps depicting intensity of ventilator support in preterm lambs.** Types of respiratory support are graded according to decreasing invasivity and intensity of support: Red—intubated and mechanically ventilated (MV); orange–intubated with continuous positive airway pressure (ET CPAP); green–nasal CPAP (nCPAP); light blue—nasal humidified high flow (nHHF) and blue—no support (none). Lambs within Sal/Sal, LPS/Sal and LPS/hAEC groups are represented in left-to-right in the order of delivery. The first 6 days of respiratory support are presented as columns, from left-to-right, as the proportion of time spend on any mode of respiratory support on that day.

**Table 3. Physiological variables in preterm lambs over the first 6 days of life.**

| Variable | Day | Experimental group | | |
| --- | --- | --- | --- | --- |
| | | Sal/Sal | LPS/Sal | LPS/hAECs |
| Proportion time on mechanical ventilation (%) | 1 | 19.9 | 44.0 | 75.7 |
| | 2 | 11.1 | 57.4 | 89.9 |
| | 3 | 10.2 | 51.8 | 79.9 |
| | 4 | 26.9 | 36.6 | 75.0 |
| | 5 | 29.2 | 33.8 | 94.4 |
| | 6 | 0.0 | 41.7 | 83.3 |
| FiO$_2$ | 1 | 0.23 ± 0.0 | 0.22 ± 0.0 | 0.22 ± 0.0 |
| | 2 | 0.21 ± 0.0 | 0.23 ± 0.0 | 0.22 ± 0.0 |
| | 3 | 0.23 ± 0.0 | 0.23 ± 0.0 | 0.23 ± 0.0 |
| | 4 | 0.23 ± 0.0 | 0.27 ± 0.1 | 0.23 ± 0.0 |
| | 5 | 0.24 ± 0.0 | 0.30 ± 0.2 | 0.24 ± 0.0 |
| | 6 | 0.24 ± 0.0 | 0.26 ± 0.1 | 0.27 ± 0.0 |
| Mean BP (mmHg) | 1 | 61 ± 2 | 62 ± 1 | 60 ± 3 |
| | 2 | 50 ± 2 | 58 ± 2 | 59 ± 1 |
| | 3 | 63 ± 3 | 60 ± 2 | 62 ± 1 |
| | 4 | 62 ± 2 | 63 ± 2 | 62 ± 2 |
| | 5 | 64 ± 2 | 63 ± 2 | 62 ± 4 |
| | 6 | 62 ± 2 | 68 ± 3 | 62 ± 3 |
| HR (beats/min) | 1 | 221 ± 17 | 212 ± 22.09 | 195 ± 17 |
| | 2 | 200 ± 16 | 191 ± 26.94 | 195 ± 31 |
| | 3 | 201 ± 28 | 190 ± 26.44 | 186 ± 31 |
| | 4 | 215 ± 30 | 186 ± 26* | 197 ± 28 |
| | 5 | 213 ± 24 | 193 ± 32 | 202 ± 23 |
| | 6 | 222 ± 22 | 208 ± 29 | 211 ± 17 |
| RR (breaths/min) | 1 | 54 ± 2 | 55 ± 2 | 51 ± 2 |
| | 2 | 58 ± 2 | 49 ± 3 | 50 ± 4 |
| | 3 | 54 ± 3 | 50 ± 2 | 47 ± 4 |
| | 4 | 54 ± 3 | 53 ± 2 | 49 ± 3 |
| | 5 | 52 ± 4 | 56 ± 3 | 53 ± 4 |
| | 6 | 61 ± 4 | 55 ± 4 | 49 ± 3* |

Data are mean ± SD.

*$P < 0.05$ compared to Sal/Sal. FiO$_2$: fraction of inspired oxygen; BP: blood pressure; HR: heart rate; RR: respiratory rate. Data analysed using 2 way repeated measures ANOVA, with time as a factor.

However, Sal/Sal lambs tended to have lower tissue areal fraction compared to LPS/Sal lambs (H(2) = 5.5; P = 0.06; mean ranks: 9.8 for Sal/Sal, 17.8 for LPS/Sal and 12.1 for LPS/hAEC). Septal crest areal fraction was not different between Sal/Sal and LPS/Sal lambs (1.8 ± 0.3 vs. 1.4 ± 0.3%) but tended to be lower in LPS/Sal lambs (3.1 ± 0.7%) than LPS/hAEC lambs (H(2) = 4.9, P = 0.08; mean ranks: 13.4 for Sal/Sal, 10.1 for LPS/Sal and 18.4 for LPS/hAEC; Fig 3).

Epithelial sloughing was not different between Sal/Sal and LPS/Sal lambs, but was lower in LPS/hAEC lambs compared to LPS/Sal (P = 0.02) and Sal/Sal groups (H(2) = 10.1, P = 0.01; mean ranks: 17.8 for Sal/Sal, 16.8 for LPS/Sal and 6.0 for LPS/hAEC; S1 Fig).

The areal density of collagen within lung tissue was not different between groups Sal/Sal: 30.8 ± 1.7%; LPS/Sal: 33.1 ± 2.6% and LPS/hAECs: 32.7 ± 2.6%; P>0.05). The areal density of

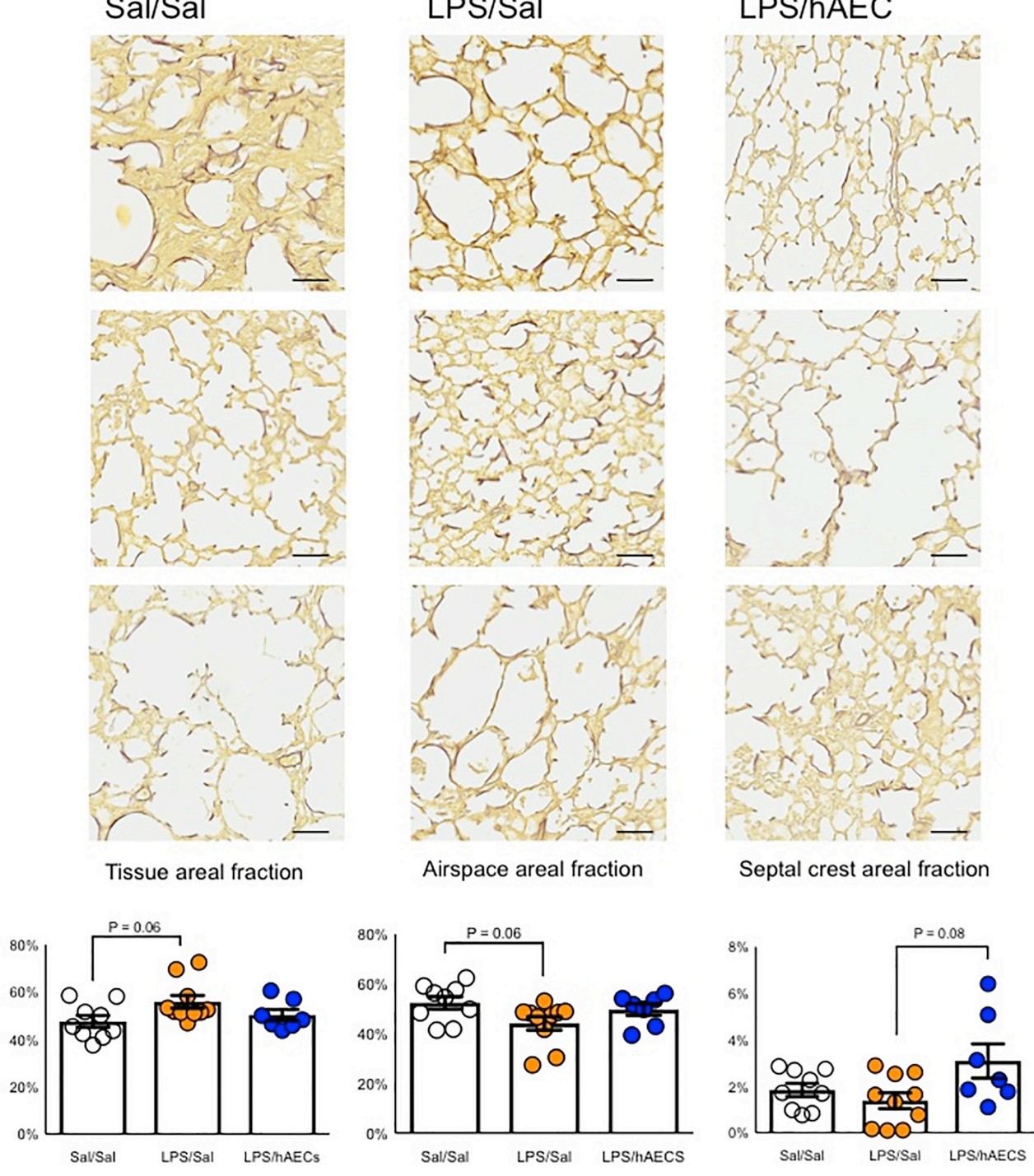

**Fig 3. Representative images depicting variability in lung parenchyma of preterm lambs within the same treatment group.** Quantification of tissue, airspace and septal crest areal fractions in the lungs of preterm lambs on day 7 of life. Elastin is stained in black and tissue in yellow. All images are taken at 20X magnification. Scale is 50 μm.

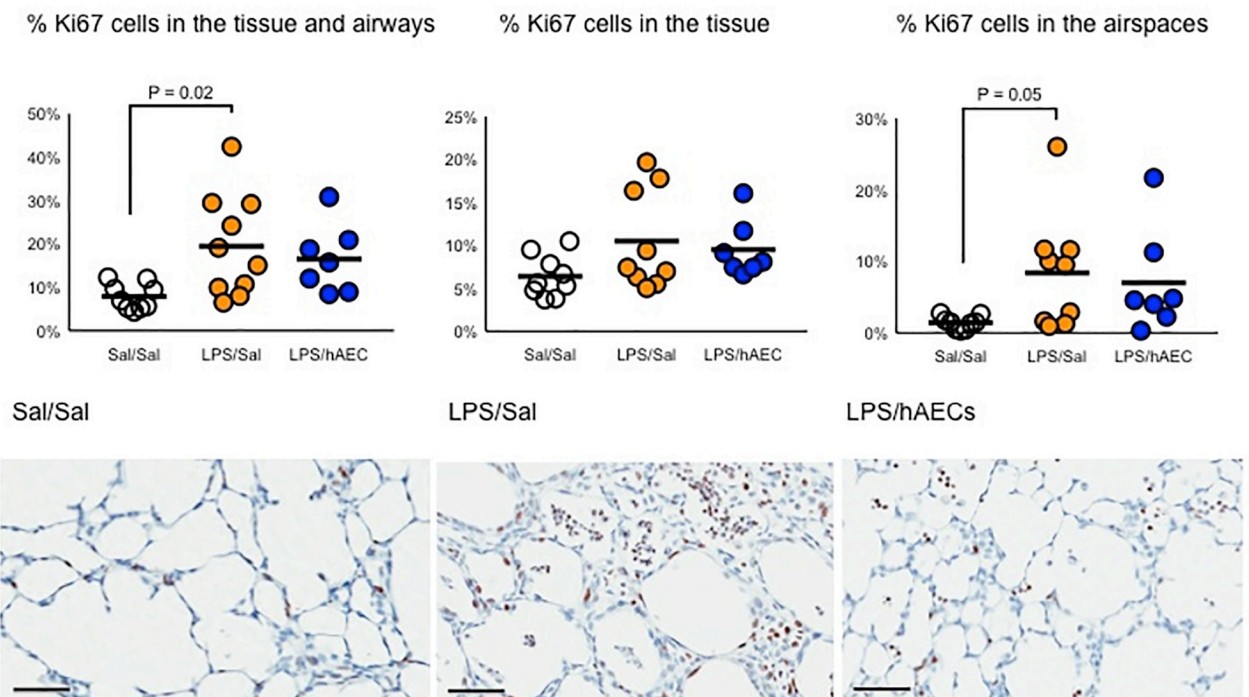

**Fig 4. Proliferating Ki67+ cells within the tissue and airspace of the lungs, within the tissue only, or within the airspaces only of preterm lambs on day 7 of life.** Representative images of the lungs of Sal/Sal, LPS/Sal and LPS/hAEC treated preterm lambs. Clumps of proliferating cells were often noted in the airspaces of animals exposed to antenatal LPS. Line represents mean. Scale is 50 μm.

elastin within lung tissue was not different between groups (Sal/Sal: 9.1 ± 0.4%; LPS/Sal: 9.2 ± 1.1% and LPS/hAECs: 9.6 ± 0.6%; P>0.99).

**Lung cell characteristics and inflammation.** The areal density of Ki67+ cells within the lung tissue and airspaces was highest in LPS/Sal lambs compared to Sal/Sal lambs (P = 0.02), and tended to be higher relative to LPS/hAEC lambs (H(2) = 10.3, P = 0.06; mean ranks: 10.7 for Sal/Sal, 19.5 for LPS/Sal and 8.6 for LPS/hAEC; Fig 4). Ki67+ cells were often clumped in the airspaces of LPS/Sal lambs (Fig 4).

Numbers of CD45+ and CD163+ cells in the lungs were not different between groups (S1 Fig). The number of pro-SP-C+ alveolar type 2 cells in the lungs were not different between treatment groups: Sal/Sal (23.6 ± 12.7 cells/ROI; LPS/Sal 19.1 ± 10.6 cells/ROI; and LPS/hAEC 20.1 ± 18.8 cells/ROI; P>0.05).

The areal density of α-SMA in the lungs was not different between Sal/Sal (38.2 ± 3.6%) and LPS/Sal (28.0 ± 6.7%) groups but tended to be lower in LPS/hAEC lambs than in the Sal/Sal group (20.1 ± 4.8%; P = 0.07).

**Pulmonary gene expression.** *IL-6*, *IL-10*, *TGF-β*, *VEGF-A*, *SP-A*, *SP-D* and *CTGF* mRNA levels were not different between Sal/Sal lambs and LPS/Sal lambs (Fig 5). *IL-1α*, *IL-1β*, *IL-8* and *TNF* mRNA levels were higher in LPS/Sal and LPS/hAEC lambs compared to Sal/Sal lambs (Fig 5).

**Hepatic gene expression.** Levels of mRNA for IL-1α, IL-1β, IL-4, IL-6, IL-8, IL-10, TNF, TGF-β, MMP-9, MMP-12, CCL2, PECAM, Hepcidin, SAA and VEGF-A were not different between Sal/Sal, LPS/Sal and LPS/hAEC lambs (S2 Table).

**In vitro studies.** The viability of hAECs was not different between isolates cultured at 33°C, 37°C or 39°C, although viability declined over 48 and 72 hours (p = 0.003 and p = 0.001 respectively) in all cultures (S2 Fig).

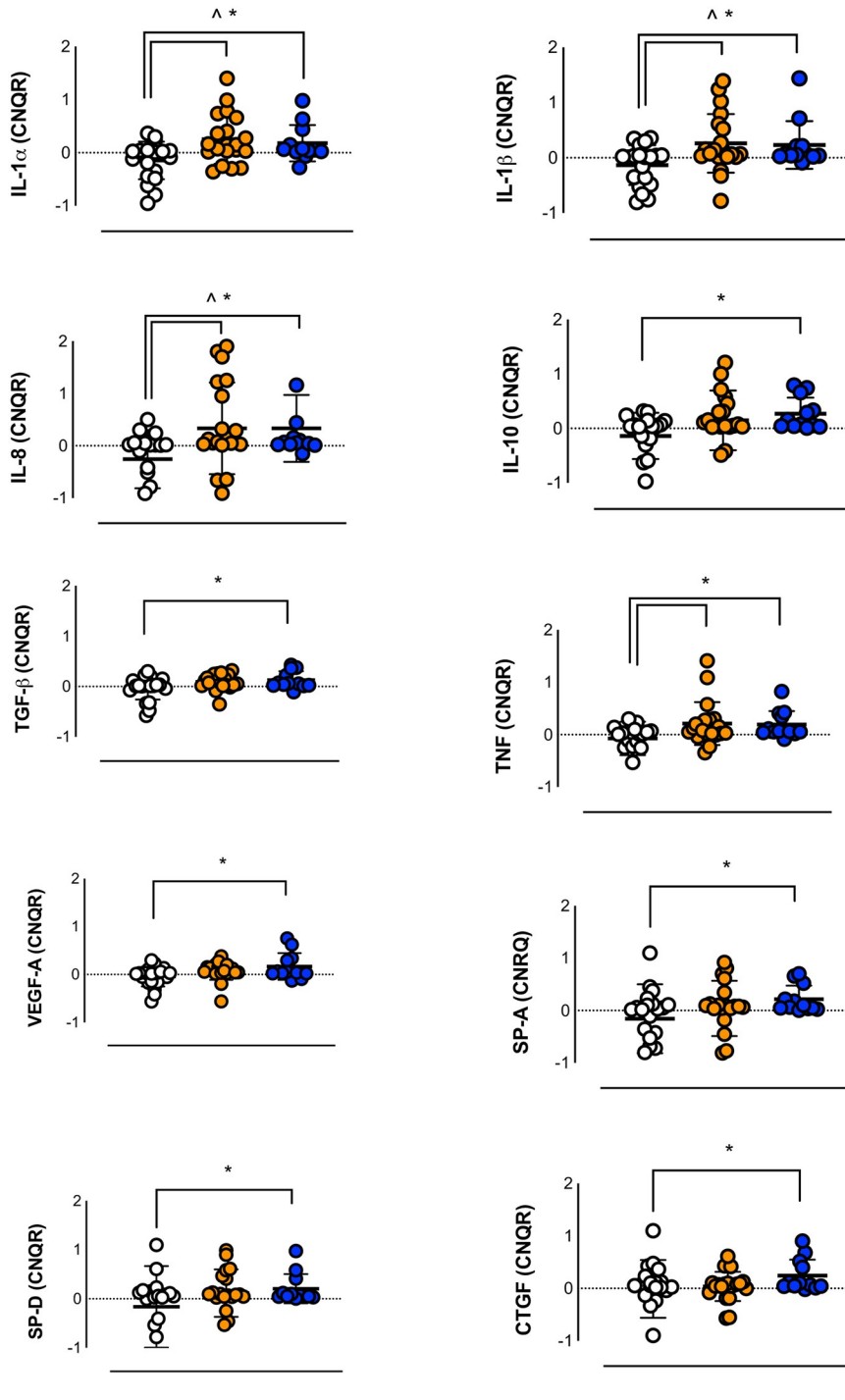

**Fig 5. mRNA expression in the lungs of Sal/Sal (clear circles), LPS/Sal (orange circles) and LPS/hAEC (blue circles) treated preterm lambs.** Data represent calibrated normalized relative quantity (CNRQ). Sal/Sal group is normalized to approximately 0.00 using qBase+ software. LPS/Sal and LPS/hAEC groups are expressed as a fold change from Sal/Sal. Open circles are Sal/Sal, yellow circles are LPS/Sal and blue circles are LPS/hAECs. *Signifies P<0.05 between Sal/Sal and Sal/hAEC. ^Signifies P<0.05 between Sal/Sal and LPS/Sal. Data are mean ± SD in duplicates.

Wound coverage was not different between hAECs cultured at 33˚C, 37˚C or 39˚C (S3 Fig). Phagocytic capacity of iMACS was evident in all cultures but was not influenced by hAEC-conditioned media from 33˚C, 37˚C or 39˚C cultures (Fig 6).

*Ureaplasma spp* were not detected in primary hAEC isolates (S4 Fig).

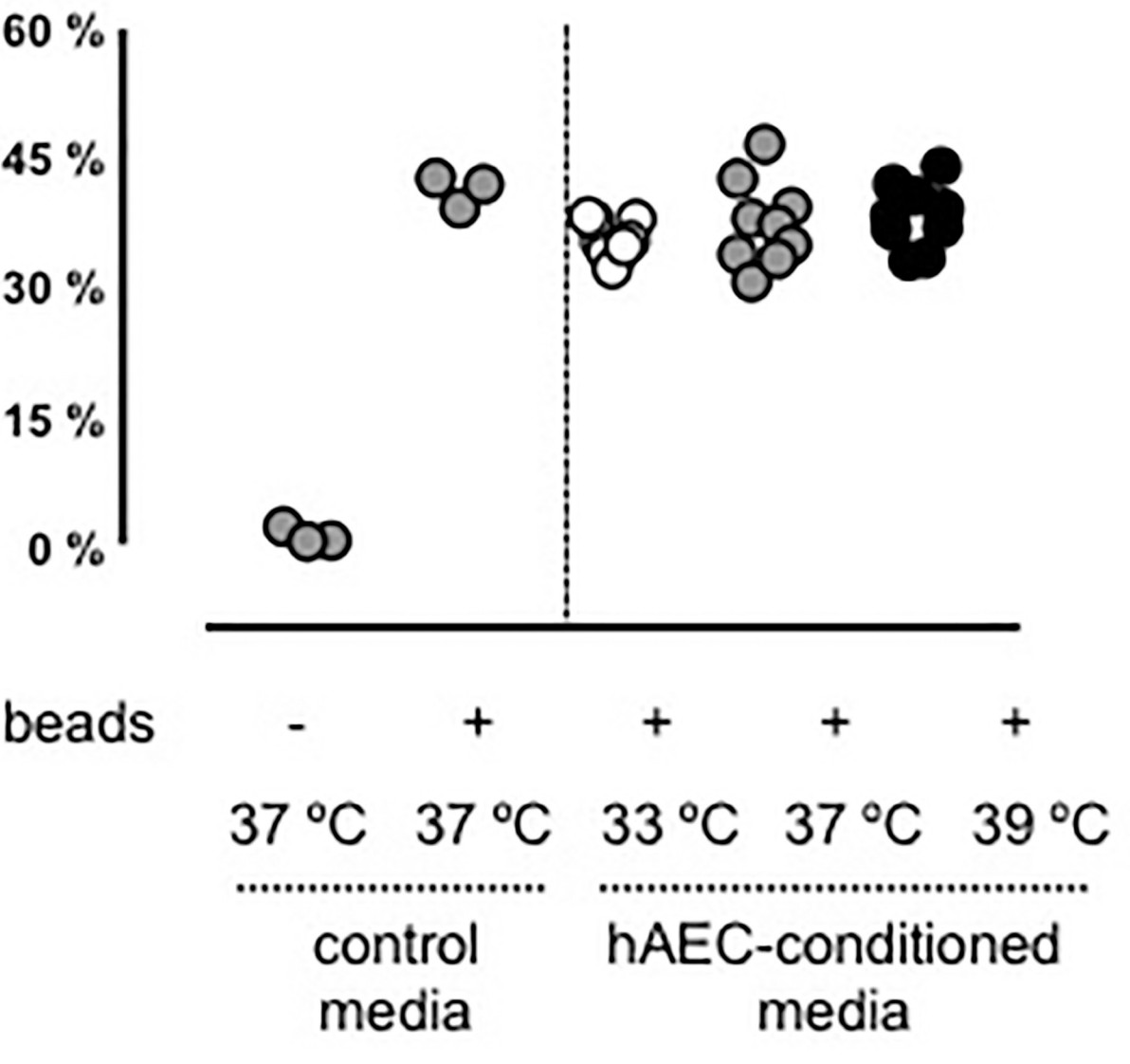

**Fig 6. hAEC conditioned media at 33˚C, 37˚C or 39˚C does not influence mouse macrophage phagocytic activity.** Immortalized mouse macrophages were incubated with standard media or with hAEC-conditioned media from temperatures 33˚C, 37˚C and 39˚C (n = 9, performed in triplicate). Percentage represents proportion of cells that phagocytosed FITC-marked fluorescent beads. Open circles are representative of hAECs cultured at 33˚C, grey circles are hAECs cultured at 37˚C and black circles are hAECs cultured at 39˚C.

## Discussion

We employ a clinically relevant model of preterm birth to investigate the impact of antenatal LPS on postnatal lung development, and additionally whether or not postnatal hAECs modulate the response to antenatal LPS.

LPS- or ventilation-induced lung injury are well characterized in preterm lambs, causing immune cell infiltration, sustained inflammation, epithelial sloughing and alveolar hypoplasia [26, 38], all of which increase the risk of BPD. The combination of IA LPS and postnatal ventilation significantly increases lung injury and inflammation in preterm lambs, compared to either alone [39].

Preterm lambs in these experiments were delivered at a gestational age that would normally result in rapid onset, fatal cardiorespiratory failure without postnatal respiratory support. We observed slight but consistent impairments in respiratory function of lambs exposed to antenatal inflammation compared to our saline treated lambs, but these functional respiratory impairments were generally manageable with usual clinical adjustments to care (e.g. manipulation of $F_IO_2$ and end-expiratory pressure). Histological indicators of lung injury, such as thickened lung parenchyma, coupled with higher proportions of proliferating cells that indicate an inflammatory response to injury, could explain the increasing requirement for $F_IO_2$ in LPS/Sal lambs. Overall, the lung structure of LPS/Sal lambs appeared immature compared to Sal/Sal animals, suggestive of delayed or early cessation of postnatal lung development.

Antenatal inflammation was associated with the presence of inflammation and injury in lung tissues of 1-week-old preterm lambs. A single intravenous dose of hAECs soon after preterm birth moderated the effects of antenatal inflammation on lung structure but did not profoundly affect gas exchange, requirements for ventilatory support, or the level of pro- or anti-inflammatory cytokine gene expression in the lungs.

Our decision to use a pragmatic de-escalating lung-protective respiratory support strategy and general postnatal support akin to contemporary clinical care, rather than pathological inflammatory insults of previous animal studies, allowed examination of the effects of hAECs in an experimental setting as close as possible to the environment of human patients in whom this therapy is entering clinical trials [40]. This translational approach was evident in heterogeneous effects within and between groups (most clearly outlined by variable lung morphology), because no lambs were treated identically. Such intra-group heterogeneity is similar to the heterogeneity seen in clinical responses of preterm newborn infants. However, the resulting variability within our datasets may have limited our ability to demonstrate formal statistical significance for individual outcomes.

Although de-escalation of respiratory support was protocolised, many lambs failed planned extubation due to inadequate respiratory drive, necessitating reintubation. Anecdotally, successful de-escalation of respiratory support was particularly difficult in lambs exposed to antenatal LPS due to inadequate respiratory drive, consistent with reported adverse effects of chorioamnionitis on brain development in preterm infants [41]. Systemic LPS causes inflammation at the brainstem, where neurons responsible for respiratory drive are located [42]. Systemic administration of IL-1 can decrease respiratory frequency and cause apneustic episodes in neonatal rats [43].

Administration of hAECs following antenatal LPS did not alter postnatal respiratory or cardiovascular physiology compared to lambs exposed to antenatal inflammation alone. The absence of an impact of hAECs on cardiorespiratory physiology is consistent with previous experiments using hAECs in fetal [11, 12, 17] and newborn sheep [14]. The increased oxygenation index observed on day 6 in hAEC-treated lambs is likely a chance finding given the

absence of intergroup differences overall for oxygenation indices and ventilator requirements over the study duration.

The observed beneficial effects of hAEC administration on lung structure in response to lung injury are consistent with previous observations in sheep and mice [14, 44, 45]. Signs of lung injury, including detachment of epithelial cells in the airspaces and remodelling of the lung parenchyma, were moderated by hAECs in our current study compared to LPS/Sal lambs. These findings are consistent with a previous shorter-term experiment in preterm lambs [14], demonstrating a reproducible protective effect of hAECs against ventilation-induced lung parenchymal injury.

Septal crest areal fractions in Sal/Sal, LPS/Sal and LPS/hAEC lambs were low compared to the expected ~ 8% septal crest areal fraction observed in naïve fetal animals of similar gestation [46]. Inhibition of septation of the developing lungs represents a failure of alveolarization [47], which is characteristic of BPD [48]. Reduction of septal crest areal fraction in our postnatal lambs is likely explained by the known adverse impact of antenatal corticosteroids [49], antenatal inflammation [50, 51], and postnatal ventilation [19] on normal septation of the preterm lungs.

The higher proportion of septal crests observed in the lungs of LPS/hAEC lambs compared to LPS/Sal lambs is consistent with previous work in fetal sheep [11] and indicates a reduction in the structural abnormalities that result from lung inflammation and injury. However, despite septal crest fraction in LPS/hAEC lambs exceeding fractions seen in Sal/Sal lambs (1.8%), this protection was only partial given the 3.1% septal crest areal fraction in the LPS/hAEC lambs remained below the abovementioned 8% observed in untreated and unventilated fetal sheep at a similar gestational age [46]. Importantly, we only gave a single prophylactic dose of hAEC soon after delivery which may have been insufficient to fully counteract the pro-inflammatory challenges faced by LPS-exposed, ventilated preterm lambs in the first week of life. A second explanation for this incomplete protection may be differences in duration of mechanical ventilation and LPS activity. Increased ventilator dependence is an observed complication in human infants exposed to chorioamnionitis [52]. By chance, a higher proportion of LPS/hAEC lambs were managed in the second year of the study using a batch of LPS that appeared anecdotally to have a greater negative effect on the respiratory drive of the preterm lambs. LPS batches may be an important consideration for all future studies involving IA LPS. That we still observed positive effects of hAECs on lung structure despite the potential detrimental impact of increased duration of respiratory support after antenatal inflammation for lambs in the second year of the study supports a therapeutic benefit of hAECs for preserving developmental potential of the lungs.

The peak inflammatory response of the fetal lungs to IA LPS occurs two days after injection [23]. Thus, at the time of delivery, preterm lambs in this study likely had maximal lung inflammation. The application of hAECs in naïve ventilated preterm lambs increases the expression of the anti-inflammatory cytokine, IL-10, consistent with our data. In mice, hAEC administration polarises macrophages from a pro-injury (M1) to a pro-reparative (M2) phenotype [53], suggesting immunomodulation is the likely mechanism of action of hAECs.

The immunomodulatory properties of hAECs likely underlie the beneficial effects of hAECs in large animal models of perinatal lung and brain injury [11, 12, 54], in which IL-1 and IL-6 expression are reduced [11, 12]. Suppression of IL-1 and TNF in mice receiving hAECs following prenatal LPS and postnatal hyperoxia supports the immunomodulatory properties of hAECs [10]. However, we did not see reduced pro-inflammatory cytokine expression in preterm lambs receiving hAEC after antenatal LPS exposure in this study, despite increased IL-10 compared to LPS/Sal lambs. One possible explanation for this discrepancy may be the additive proinflammatory effects of antenatal LPS and postnatal respiratory

management, causing a heightened immune response in preterm lambs (there were no differences in inflammatory gene expression between LPS/Sal and LPS/hAEC lambs at 7 days of age). Another possible explanation for this discrepancy is the number of hAECs used in this study (30 million), which was lower than that previously used in fetal sheep (90–180 million) [11, 12] and preterm lambs (180 million) [14], although higher than that used in a Phase I (safety) trial of hAECs (1 million cells/kg) in preterm infants with established BPD [55]. Our cell dose is consistent with that of a planned Phase II dose-escalation trial that will administer between 2–30 million hAECs/kg to preterm infants at risk of BPD [40]. Our decision to deliver hAECs intravenously is similarly consistent with clinical trials. Previous studies show no difference in intravenous or intratracheal administration of hAECs in fetal sheep [12].

*In vitro* experiments performed as part of this study are reassuring with respect to the use of hAECs in preterm infants. Indices of hAEC viability and function *in vitro* were not influenced by variations in temperature ranging from 33–39˚C, a realistic range for preterm infants who might undergo therapeutic hypothermia [56] or who are at risk of fever associated with sepsis [57]. The validity of results from previous animal experiments using species with varying body temperatures (normothermia is 36˚C or 39˚C for mice or sheep, respectively and amniotic fluid temperature is ~39˚C in pregnant ewes) is also reassured by our *in vitro* results.

## Limitations and considerations

A pragmatic approach to design and execution of experiments such as these is a necessary, but potentially confounding reality. The requirement for 24-hour neonatal care places a limit on contemporaneous experimentation of preterm lambs, which necessitated the performance of experimentation over a period of more than 12 months. While clinical treatment of the postnatal lambs did not change appreciably over the two study years, seasonality of sheep reproduction and peri-conceptional/prenatal nutrition may have increased variability of observed responses. However, perhaps more importantly, we also needed to use two different batches of LPS and in retrospect, these two LPS batches appeared to have different clinical impacts on the respiratory drive of the preterm lambs, suggesting different LPS activity. The unbalanced assignment of lambs to groups between the two LPS batches may have biased against the LPS/hAEC group, which received the more "active" LPS preparation. Although use of ANCOVA for statistical analysis [58] allowed adjustment for this confounding variable, we cannot exclude an effect of LPS batch on outcomes.

## Conclusion

Antenatal inflammation was associated with lung inflammation and injury in 1-week-old preterm lambs. A single intravenous dose of hAECs soon after preterm birth moderated the effects of antenatal inflammation on lung structure but did not profoundly affect gas exchange, requirements for ventilatory support, or the level of pro- or anti- inflammatory cytokine gene expression in the lungs. Impairment of alveolarization, considered a hallmark of BPD, may be moderated by hAECs. Further experimentation to determine the effects of hAECs in preterm neonates will guide implementation of this therapy in preterm human infants.

## Supporting information

**S1 Fig. The number of CD45+ and CD163+ cells and epithelial sloughing scores in the lungs of Sal/Sal, LPS/Sal and LPS/hAEC preterm lambs on day 7 of life.** Horizontal lines show group mean.
(DOCX)

**S2 Fig. hAECs apoptotic activity is influenced by time, not culture temperatures of 33˚C, 37˚C or 39˚C.** (A) The proportion of cells positive for Annexin V increased with time, irrespective of culture temperature (n = 9, performed in triplicate). (B) hAECs positive for 7AAD did not change with culture temperature or time (n = 9, performed in triplicate). Open circles are representative of hAECs cultured at 33˚C, grey circles are hAECs cultured at 37˚C and black circles are hAECs cultured at 39˚C.
(DOCX)

**S3 Fig. hAECs wound healing capacity is the same in 33˚C, 37˚C or 39˚C cultures.** (A) Percentage change in the wound area recovered by hAECs exposed to culture temperatures 33˚C, 37˚C and 39˚C. (B) Representative images of cells cultured at 33˚C, 37˚C and 39˚C, at 0 and 72 hours (n = 9, performed in triplicate). Open circles are representative of hAECs cultured at 33˚C, grey circles are hAECs cultured at 37˚C and black circles are hAECs cultured at 39˚C.
(DOCX)

**S4 Fig. Electrophoretic analysis of PCR ureaplasma products from 19 hAEC donors.** M: Molecular sized strands. Lane 20 is positive control. Lane 21 is a negative control.
(DOCX)

**S1 Table. Probe selection for TaqMan® assays.**
(DOCX)

**S2 Table. Fold change mRNA expression of genes in the liver of preterm lambs.**
(DOCX)

**S1 File. Supplementary methodology.** Detailed methodology outlining animal procedures and the general care of preterm lambs.
(DOCX)

**S1 Raw image. M: Molecular sized strands.** Lane 20 is positive control. Lane 21 is a negative control.
(PDF)

## Author Contributions

**Conceptualization:** Rebecca Lim, Euan Wallace, J. Jane Pillow, Timothy J. Moss.

**Data curation:** Paris Clarice Papagianis, J. Jane Pillow, Timothy J. Moss.

**Formal analysis:** Graeme Polglase, J. Jane Pillow, Timothy J. Moss.

**Funding acquisition:** Rebecca Lim, Euan Wallace, Graeme Polglase, J. Jane Pillow, Timothy J. Moss.

**Investigation:** Paris Clarice Papagianis, J. Jane Pillow, Timothy J. Moss.

**Methodology:** Graeme Polglase, J. Jane Pillow, Timothy J. Moss.

**Project administration:** Paris Clarice Papagianis, Siavash Ahmadi-Noorbakhsh, J. Jane Pillow, Timothy J. Moss.

**Resources:** Rebecca Lim, Graeme Polglase, J. Jane Pillow, Timothy J. Moss.

**Software:** J. Jane Pillow, Timothy J. Moss.

**Supervision:** Graeme Polglase, J. Jane Pillow, Timothy J. Moss.

**Validation:** J. Jane Pillow, Timothy J. Moss.

**Visualization:** J. Jane Pillow, Timothy J. Moss.

**Writing – original draft:** Paris Clarice Papagianis, J. Jane Pillow, Timothy J. Moss.

**Writing – review & editing:** Paris Clarice Papagianis, Siavash Ahmadi-Noorbakhsh, Rebecca Lim, Euan Wallace, Graeme Polglase, J. Jane Pillow, Timothy J. Moss.

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
