## [Decision Letter · Decision Letter 0]

24 Mar 2021

PONE-D-21-04396

The effect of human amnion epithelial cells on lung development and inflammation in preterm lambs exposed to antenatal inflammation

PLOS ONE

Dear Dr. Papagianis,

Thank you for submitting your manuscript to PLOS ONE. After careful consideration, we feel that it has merit but does not fully meet PLOS ONE’s publication criteria as it currently stands. Therefore, we invite you to submit a revised version of the manuscript that addresses the points raised during the review process.

I kindly ask you to address all  specific points raised by both reviewers. Their helpful comments can help you to specify some aspects of data presentation within the manuscript.

We look forward to receiving your revised manuscript.

Kind regards,

Harald Ehrhardt

Academic Editor

PLOS ONE

Journal Requirements:

3. Please include your tables as part of your main manuscript and remove the individual files. Please note that supplementary tables should remain as separate "supporting information" files.

6.  Thank you for stating the following in the Financial Disclosure section:

'This research was supported by an NHMRC Project Grant (1077769), NHMRC Centre for Research Excellence (1057514), two NHMRC Senior Research Fellowships (JJP; 1077691: TJM 1043294), the Victorian Government’s Operational Infrastructure Support Program, and the West Australian Government’s Medical and Health Research Infrastructure Fund. Unrestricted equipment and consumable support was provided by Chiesi Farmaceutici S.p.A. (poractant alfa); Fisher & Paykel Healthcare (ventilator circuits); and ICU Medical (arterial monitoring lines).'

We note that you received funding from a commercial sources: Chiesi Farmaceutici S.p.A., Fisher & Paykel Healthcare and ICU Medical.

a. Please provide an amended Competing Interests Statement that explicitly states thes commercial funders, along with any other relevant declarations relating to employment, consultancy, patents, products in development, marketed products, etc.

b. Please also state what role the funders took in the study.  If the funders had no role, please state: "The funders had no role in study design, data collection and analysis, decision to publish, or preparation of the manuscript."

If this statement is not correct you must amend it as needed..

c. Please include your amended Competing Interests Statement and amended Role of Funder statement within your cover letter. We will change the online submission form on your behalf.

Reviewers' comments:

Reviewer's Responses to Questions

**Comments to the Author**

1. Is the manuscript technically sound, and do the data support the conclusions?

Reviewer #1: No

Reviewer #2: Yes

2. Has the statistical analysis been performed appropriately and rigorously? 

Reviewer #1: Yes

Reviewer #2: Yes

3. Have the authors made all data underlying the findings in their manuscript fully available?

Reviewer #1: Yes

Reviewer #2: Yes

4. Is the manuscript presented in an intelligible fashion and written in standard English?

Reviewer #1: Yes

Reviewer #2: Yes

5. Review Comments to the Author

Reviewer #1: Comments to the Author:

The authors hypothesized that human amnion epithelial cells (hAECs) would reduce lung injury in preterm lambs-exposed to antenatal inflammation and investigated the effects of a single dose of intravenous hAECs on lung development and respiratory requirement. The authors found that hAECs administration at birth did not reduce respiratory requirements or lung development in preterm lambs exposed to antenatal inflammation. This study has many data and the results do not confirm the hypothesis and the connection between in vitro and animal study was not evident.

General concerns:

1. This study presents many data and most data did not show clear evidence of positive effects.

2. The conclusions in Abstract was not supported by the results: Postnatal administration of a single dose of hAECs activates the pulmonary immune system without changing ventilator requirements in preterm lambs born after intrauterine inflammation. However, activation of pulmonary immune system was not described in the manuscript.

3. Methods: Please describe how the cell numbers of hAECs (30 � 106) was decided in this study.

4. Please describe and discuss the rationale to measure hepatic gene expression in lung injury model.

5. Methods: The animal numbers were not consistent. (LPS/Sal, n=10; or Sal/Sal, n=9) in Abstract. Preterm lamb studies on page 7, Ultrasound-guided intra-amniotic (IA) injection of lipopolysaccharide (LPS;4 mg; 2 mg/mL; Escherichia coli 055:B5; Sigma-Aldrich, NSW, Australia; n=10) or saline (n=10) was performed at 126 days’ GA.

6. Methods: The methods to collect the Physiology variables in Table 2 were not described.

7. Results: Please insert (Table 1) in the third paragraph on page 15.

8. Figure Legend 1: “^Signifies P<0.05 between Sal/Sal and LPS/hAECs” was not shown in the Figure 1. Please describe the meaning of **.

Reviewer #2: Thank you for submitting the manuscript for review.

The study highlights the role of inflammatory axis in the genesis of CLD. By and large the manuscript reads well. I would like to add couple of comments for minor revision

1) Explicitly state the three groups based on the antenatal- post natal interventions - e.g. group A (LPS/S); Group B (S/S) to improve clarity

2) The abstract, discussion, conclusions- explicitly state the effect of hAEC on inflammatory cytokines- while it states hAEC influence pulmonary immunology -- for better clarity state the effect of hAEC on pro and anti-inflammatory cytokine response. This aspect is also discussed appropriately in the limitations.

6. PLOS authors have the option to publish the peer review history of their article (what does this mean?). If published, this will include your full peer review and any attached files.

Reviewer #1: **Yes: **Chung-Ming Chen MD, PhD.

Reviewer #2: **Yes: **Arun Sasi

---

## [Author Response · Author response to Decision Letter 0]

12 May 2021

Dear Editor,

Thank you for your consideration of our manuscript “The effect of human amnion epithelial cells on lung development and inflammation in preterm lambs exposed to antenatal inflammation”. We believe we have adjusted this manuscript to meet the journal requirements and comments from the editor and reviewers 1 and 2. Please note that our original blot image is attached as a separate PDF file “S4_raw_image” and as part of supplementary figures (S4 Fig.). 

Please find our responses to individual reviewers below. We have attached a marked-up version of the manuscript with changes in red text. We have addressed concerns of funding bodies raised by the editor in text on p. 33, line starting 702.

Additionally, please note that we inadvertently omitted one author, Dr. Graeme Polglase. We recognise the addition of this author and this author has reviewed the manuscript. Polglase’s contributions to the manuscript are outlined in red on p.ii, line 27-8. All authors support the alterations and submission of this manuscript.

We greatly appreciate your consideration of this manuscript.

Kind Regards,

Paris Papagianis and co-authors.

Reviewer #1:

The authors hypothesized that human amnion epithelial cells (hAECs) would reduce lung injury in preterm lambs-exposed to antenatal inflammation and investigated the effects of a single dose of intravenous hAECs on lung development and respiratory requirement. The authors found that hAECs administration at birth did not reduce respiratory requirements or lung development in preterm lambs exposed to antenatal inflammation. This study has many data and the results do not confirm the hypothesis and the connection between in vitro and animal study was not evident.

General concerns:

1. This study presents many data and most data did not show clear evidence of positive effects.

We believe that the data presented provide important insight into the use of translational studies investigating cellular therapies for lung disease. We use a clinically relevant preterm intensive care model which recapitulates human practice to demonstrate that while the impact of hAECs on ventilator requirements is minimal the positive effects on lung parenchyma are apparent. Additionally, we analyse the impact of contemporary NICU care in the context of chorioamnionitis (a common precursor to preterm birth and NICU). The outcomes of these studies are informing the planning and design of clinical trials using hAECs in human infants. 

2. The conclusions in Abstract was not supported by the results: Postnatal administration of a single dose of hAECs activates the pulmonary immune system without changing ventilator requirements in preterm lambs born after intrauterine inflammation. However, activation of pulmonary immune system was not described in the manuscript.

We describe markers of pulmonary inflammation in Fig. 5. We have removed the word “activation” from the abstract and replaced with “stimulates a pulmonary immune response”.

3. Methods: Please describe how the cell numbers of hAECs (30 � 106) was decided in this study.

We chose a hAEC dose of roughly 10 million cells per kilo, based on the average weight of preterm lambs at 128 dGA (~3kg). This dosing approach for hAECs was in line with a planned Phase 2 clinical trial for hAECs in preterm infants with BPD when our study was conducted.

4. Please describe and discuss the rationale to measure hepatic gene expression in lung injury model.

Although the focus of this study was on ventilator requirements and lung development/maturation, we opportunistically looked at markers for systemic inflammation. This analysis of hepatic gene expression as an indicator of systemic inflammation informed our conclusion that hAECs were administered safely in our preterm lambs.

5. Methods: The animal numbers were not consistent. (LPS/Sal, n=10; or Sal/Sal, n=9) in Abstract. Preterm lamb studies on page 7, Ultrasound-guided intra-amniotic (IA) injection of lipopolysaccharide (LPS;4 mg; 2 mg/mL; Escherichia coli 055:B5; Sigma-Aldrich, NSW, Australia; n=10) or saline (n=10) was performed at 126 days’ GA.

Thank you. Unfortunately, this is typological error in the abstract. Saline/Saline = 9. This group size number has been amended for the Saline/Saline group throughout the manuscript. 

6. Methods: The methods to collect the Physiology variables in Table 2 were not described.

All the methodology used to collect physiological variables are described in detail in the supplementary methods due to word restrictions in the main paper. A detailed methodology for data collection of physiological variables is cited in the general postnatal management section of our methodology in the main manuscript (line 152 onwards). 

7. Results: Please insert (Table 1) in the third paragraph on page 15.

Please find Table 1 at line 242

8. Figure Legend 1: “^Signifies P<0.05 between Sal/Sal and LPS/hAECs” was not shown in the Figure 1. Please describe the meaning of **.

Figure legends are now adjusted to define all symbols in each figure. ^Signifies P<0.05 between Sal/Sal and LPS/Sal. *Signifies P<0.05 between Sal/Sal and LPS/hAECs &Time signifies P<0.05 change over 1-6 days, not between treatment groups.” There is no use of ** in any of the figures, and these symbols are consistent for all subsequent figures. Additionally, we have simplified the description of the colour coding in figure 2 and enhanced the labelling in Figures 1 and 5 to improve readability.

Reviewer #2: 

The study highlights the role of inflammatory axis in the genesis of CLD. By and large the manuscript reads well. I would like to add couple of comments for minor revision

1) Explicitly state the three groups based on the antenatal- post natal interventions - e.g. group A (LPS/S); Group B (S/S) to improve clarity

The use of A, B and C for treatment groups may be more confusing as the reader has to remember what each represents. We abbreviate Saline to Sal and human amnion epithelial cells to hAEC, as is standard within the literature. 

2) The abstract, discussion, conclusions- explicitly state the effect of hAEC on inflammatory cytokines- while it states hAEC influence pulmonary immunology -- for better clarity state the effect of hAEC on pro and anti-inflammatory cytokine response. This aspect is also discussed appropriately in the limitations

We have adjusted the abstract as per reviewer #1 comment which we believe also meets the requirements of this comment. The discussion and conclusion have also been changed accordingly. The discussion:

• Line 398 “level of pro- or anti- inflammatory cytokine gene expression in the lungs”

• Line 463: “increases the expression of the anti-inflammatory cytokine, IL-10”

• Line 471-3: “However, we did not see reduced pro-inflammatory cytokine expression in preterm lambs receiving hAEC after antenatal LPS exposure in this study, despite increased IL-10 compared to LPS/Sal lambs.”

The conclusion has been altered: “level of pro- or anti- inflammatory cytokine gene expression in the lungs”

---

## [Decision Letter · Decision Letter 1]

7 Jun 2021

The effect of human amnion epithelial cells on lung development and inflammation in preterm lambs exposed to antenatal inflammation

PONE-D-21-04396R1

Dear Dr. Papagianis,

We’re pleased to inform you that your manuscript has been judged scientifically suitable for publication and will be formally accepted for publication once it meets all outstanding technical requirements.

Kind regards,

Harald Ehrhardt

Academic Editor

PLOS ONE

Additional Editor Comments (optional):

Reviewers' comments:

Reviewer's Responses to Questions

**Comments to the Author**

1. If the authors have adequately addressed your comments raised in a previous round of review and you feel that this manuscript is now acceptable for publication, you may indicate that here to bypass the “Comments to the Author” section, enter your conflict of interest statement in the “Confidential to Editor” section, and submit your "Accept" recommendation.

Reviewer #1: All comments have been addressed

2. Is the manuscript technically sound, and do the data support the conclusions?

Reviewer #1: Yes

3. Has the statistical analysis been performed appropriately and rigorously? 

Reviewer #1: Yes

4. Have the authors made all data underlying the findings in their manuscript fully available?

Reviewer #1: Yes

5. Is the manuscript presented in an intelligible fashion and written in standard English?

Reviewer #1: Yes

6. Review Comments to the Author

Reviewer #1: The authors have addressed the concerns. I have no further concerns.

The authors have addressed the concerns.

7. PLOS authors have the option to publish the peer review history of their article (what does this mean?). If published, this will include your full peer review and any attached files.

Reviewer #1: **Yes: **Chung-Ming Chen

---

## [Editor Report · Acceptance letter]

16 Jun 2021

PONE-D-21-04396R1 

The effect of human amnion epithelial cells on lung development and inflammation in preterm lambs exposed to antenatal inflammation 

Dear Dr. Papagianis:

I'm pleased to inform you that your manuscript has been deemed suitable for publication in PLOS ONE. Congratulations! Your manuscript is now with our production department. 

Kind regards, 

on behalf of

Prof. Harald Ehrhardt 

Academic Editor

PLOS ONE